# Gut bacteria influence *Blastocystis* sp. phenotypes and may trigger pathogenicity

**Arutchelvan Rajamanikam**[1]*, **Mohd Noor Mat Isa**[2], **Chandramathi Samudi**[3]*, **Sridevi Devaraj**[4]*, **Suresh Kumar Govind**[1]*

1 Department of Parasitology, Faculty of Medicine, University Malaya, Kuala Lumpur, Malaysia, 2 Malaysian Genome and Vaccine Institute, Jalan Bangi, Kajang, Malaysia, 3 Department of Medical Microbiology, Faculty of Medicine, University Malaya, Kuala Lumpur, Malaysia, 4 Texas Children's Microbiome Center, Houston, Texas, United States of America

\* arun04@um.edu.my (AR); chandramathi@um.edu.my (CS); sxdevara@texaschildrens.org (SD); suresh@um.edu.my (SKG)

**Data Availability Statement:** All the data generated are included within the manuscript. The demographic data of the participants have been included in the Supporting Information files. The

## Abstract

Whilst the influence of intestinal microbiota has been shown in many diseases such as irritable bowel syndrome, colorectal cancer, and aging, investigations are still scarce on its role in altering the nature of other infective organisms. Here we studied the association and interaction of *Blastocystis* sp. and human intestinal microbiota. In this study, we investigated the gut microbiome of *Blastocystis* sp.-free and *Blastocystis* sp. ST3-infected individuals who are symptomatic and asymptomatic. We tested if the expression of phenotype and pathogenic characteristics of *Blastocystis* sp. ST3 was influenced by the alteration of its accompanying microbiota. *Blastocystis* sp. ST3 infection alters bacterial composition. Its presence in asymptomatic individuals showed a significant effect on microbial richness compared to symptomatic ones. Inferred metagenomic findings suggest that colonization of *Blastocystis* sp. ST3 could contribute to the alteration of microbial functions. For the first time, we demonstrate the influence of bacteria on *Blastocystis* sp. pathogenicity. When *Blastocystis* sp. isolated from a symptomatic individual was co-cultured with bacterial suspension of *Blastocystis* sp. from an asymptomatic individual, the parasite demonstrated increased growth and reduced potential pathogenic expressions. This study also reveals that *Blastocystis* sp. infection could influence microbial functions without much effect on the microbiota diversity itself. Our results also demonstrate evidence on the influential role of gut microbiota in altering the characteristics of the parasite, which becomes the basis for the contradictory findings on the parasite's pathogenic role seen across different studies. Our study provides evidence that asymptomatic *Blastocystis* sp. in a human gut can be triggered to show pathogenic characteristics when influenced by the intestinal microbiota.

## Author summary

Single-cell eukaryotes in the intestinal microbiota are increasingly gaining attention for their ability to influence microbiota composition. *Blastocystis* sp. is no exception. This study for the first time demonstrates gut microbiota alteration due to the colonization of

raw sequencing microbiome data have been added to a public repository, the National Library of Medicine (NCBI) as a BioProject with accession number PRJNA88178.

**Funding:** This study was supported by Transdisciplinary Research Grant Scheme, Ministry of Higher Education (TRGS) (TRGS/1/2018/UM/01/7/1) obtained by SKG. The funders had no role in study design, data collection and analysis, decision to publish, or preparation of the manuscript.

**Competing interests:** The authors have declared that no competing interests exist.

*Blastocystis* sp. ST 3 in symptomatic and asymptomatic conditions. Colonization of *Blastocystis* sp. ST3, regardless of symptoms, significantly alters the diversity and microbiota composition. The abundance of *Prevotella* sp. was significantly elevated in symptomatic *Blastocystis* sp. carriage. Inferred metagenomic findings revealed that predicted metabolic functions were altered in *Blastocystis* sp. carriage in symptomatic and asymptomatic conditions. The current study demonstrates a bidirectional influence that seems to be crucial in *Blastocystis* sp.–microbiota interaction. Altering the accompanying microbiota of a symptomatic *Blastocystis* sp. with bacterial suspension from an asymptomatic condition resulted in the protozoan exhibiting asymptomatic characteristics. This implies, for the first time the effect of accompanying microbiota on *Blastocystis* sp. phenotypic characteristics. The findings lead to a postulation that a harmless protozoan parasite can be turned harmful by its accompanying microbiota in the gut.

## Introduction

The gut microbiota is composed of not only prokaryotes; but also, certain eukaryotes, most notably, the intestinal protozoans that pose a serious health burden in developing countries. Recent studies suggest that although the gut microbiota is diverse in species, the temporal fluctuation of certain microbial species (denotes instability) commonly occurs [1,2]. Whether this fluctuation in gut microbiota exerts an influence on other eukaryotic inhabitants of the gut is not much explored.

*Blastocystis* sp. is an intestinal protozoan parasite that has been frequently associated with general gastrointestinal symptoms, colorectal cancer (CRC), and irritable bowel disease (IBS) [3]. Despite its high prevalence, *Blastocystis* sp. has unresolved controversies regarding its pathogenicity. It was considered as a harmless commensal with recent findings implying that there is a possibility of the existence of two variant forms i.e. disease (pathogenic) and non-disease (non-pathogenic) causing types in a single subtype. *Blastocystis* sp. has been reported to exhibit strong interaction with its accompanying microbiota [4]. Studies have reported on increased diversity of bacteria in *Blastocystis* sp. colonized gut [5,6] and strong association of specific subtype with gut microbial composition [7]. However, there is a paucity of studies on the influence of gut microbiota by a single and most prevalent subtype (ST3) of *Blastocystis* sp. isolated from symptomatic and asymptomatic individuals as well as the influence of varying accompanying microbiota on this intestinal protozoan cells.

Some of the common intestinal protozoan parasites that are pathogenic to human gut include *Entamoeba histolytica* and *Giardia duodenalis*. According to epidemiological data, colonization of these eukaryotic pathogens may not necessarily result in the manifestation of symptoms [8]. Some studies have posited that the association of certain bacteria resulted in increased pathogenicity and protective effect in the infection of *Giardia* sp. [9–11] and increased virulence in *Entamoeba histolytica* [12,13]. It is unknown if a similar mechanism occurs with *Blastocystis* sp. infection. Whether the microbial environment could influence shaping the parasite's characteristics, thus altering the severity of infection is still a question.

In this study, we investigated the gut microbiota profiles in symptomatic and asymptomatic individuals with or without *Blastocystis* sp. ST3 infection. Subsequently, we altered the microbiome of *Blastocystis* sp. in *in vitro* culture to understand the response of the parasitic cell towards different microbiota.

## Methods

### Ethics statement

A verbal and written consent was obtained from all participants recruited. The study procedure was approved by University Malaya Medical Centre (UMMC) Medical Research Ethics Committee (MRECID: 201914–6975).

### Stool sample collection and *Blastocystis* sp. carriage assignment

A total of 50 fecal samples were studied, with one fecal sample collected each from 50 individuals who participated. Twenty-eight individuals who did not experience any gastrointestinal illness were grouped as asymptomatic. These individuals were recruited from a voluntary stool survey. The balance of 22 was patients visiting the Gastroenterology Unit of University Malaya Medical Centre (UMMC) and Gastroenterology and Hepatology Specialist Clinic of Pantai Medical Centre, Kuala Lumpur, Malaysia. These patients, who experienced frequent non-specific gastrointestinal symptoms such as bloating, abdominal cramps, loose stool, and diarrhea at the time of recruitment were grouped as symptomatic. Both the symptomatic and asymptomatic individuals were sub-grouped into *Blasotcystis* sp.-infected and *Blastocystis* sp.-free groups. Only participants with *Blastocystis* sp. as the sole infective agent were recruited (for all *Blastocystis* sp.-infected individuals). For all the *Blastocystis* sp.-free individuals, the fecal specimens were screened to ensure no other parasitic infection was detected. Screening for other parasites was done using the formal-ether concentration technique as described previously [14]. Patients diagnosed with colorectal cancer (CRC), inflammatory bowel disease (IBD), or irritable bowel syndrome (IBS) and those who have consumed antibiotics within the last 30 days were excluded from participating. The clinicians obtained prior written consent before the recruitment of participants. The clinicians also confirmed the diagnosis of these individuals after a thorough examination. To maintain homogeneity in population and environment, only individuals from the most developed part of Malaysia (Kuala Lumpur) and who belonged to a high socio-economic group were included. This study was approved by the Medical Research Ethics Committee of UMMC (201914–6975). Stool samples collected in screw-capped containers were processed and stored at -20˚C within 6 hours of collection.

### Fecal DNA extraction

DNA extraction from fecal materials was carried out using MACHEREY-NAGEL NucleoSpin Soil kit (MACHEREY-NAGEL GmbH & Co. KG, Dü ren, Germany). The SL2 buffer was used with Enhancer SX in the lysis step of extraction. DNA was eluted in a final volume of 50 μl and stored at -80˚C.

### Patient and public involvement

Patients and the public were first involved during fecal sample collection and questionnaire administration. Recruited individuals were either identified by the healthcare professional or upon voluntary admission. All research questions and outcome measures were approved by the Medical Research Ethics Committee of UMMC and were explained to each participant in detail by the enumerators and the clinicians. There was no involvement of the patient or the public in the design of this study. The participants agreed verbally to have their results published.

### Amplification of variable 3 (V3) and variable 4 (V4) region of 16S ribosomal RNA (rRNA) genes

The primer pair sequences that produced a single amplicon for V3 and V4 region of approximately 460 bp were used as described [15]. The library preparation was done according to the study. The amplification process was carried out using the 2X KAPA HiFi HotStart Ready Mix with microbial genomic DNA at a concentration of 5ng/μl in 10mM. The amplification was carried out with thermal cycling consisting of 95˚C for 3 minutes, followed by 25 cycles of 95˚C (30 seconds), 55˚C (30 seconds), and 72˚C (30 seconds) with a final extension of 72˚C for 5 minutes. Subsequently, a dual index barcode Illumina sequencing adaptor was attached to the amplicon using Nextera XT Index Kit (Illumina). Prepared amplicons were cleaned up again using AMPure beads. The V3 and V4 regions were then sequenced on an Illumina MiSeq platform (Illumina, San Diego CA, USA) at the Texas Children's Microbiome Center.

### Quality filtering and analysis of filtered reads

The raw sequences were joined at the paired-end by trimming the low-quality bases. All sequences were imported and analyzed through Quantitative Insights Into Microbial Ecology 2 (QIIME 2 Version 2020.6) platform [16]. The paired-end sequences were joined, chimeric sequences filtered and low-quality reads were removed using DADA2 plug-in [17].

### Analysis of filtered reads sequence diversity

Quality filtered reads were used as the sequence data. The Operational Taxonomic Units (OTU) abundance was identified and quantified with reference to the Greengenes reference database version 13_5 with a 97% homology cut-off (https://greengenes.secondgenome.com/). We generated a phylogenetic tree by first performing multiple sequence alignment of the sequence using the mafft [18]. Next, highly variable regions that add noise to the tree were removed and the phylogenetic tree was built using FastTree [19]. This pipeline was carried out in QIIME2.

The alpha diversity was measured using the Shannon index metrics, richness and observed OTUs. Alpha rarefaction curve was plotted after samples were rarefied to 4000 sequences per sample. Beta diversity was computed using Bray-Curtis distance matrix ordinated using nonmetric multidimensional scaling (NMDS) and Principal Coordinate Analysis (PCoA).

### Linear Discriminant analysis effect size (LEfSe) and predicting the functional composition of a metagenome

LEfSe algorithm from the Galaxy web application (https://huttenhower.sph.harvard.edu/galaxy/) was used to identify taxa/gene/pathways with differential abundance from different experimental classes. In this study, we have used the symptomatic and asymptomatic groups as main classes and *Blastocystis* sp. infection status as subclasses. LEfSe lists the taxa that are differential among the classes with statistical and biological significance and ranks them according to effect size [20].

Predicted functional metagenome was developed using Phylogenetic Investigation of Communities by Reconstruction of Unobserved States (PICRUSt). An OTU table was built using closed-reference clustering method by comparing each OTU representative to the Greengenes database version 13_5 at 97% cutoff. The resulting OTU table was used for metagenome prediction in Galaxy web application using KEGG orthology classification schemes [21]. Subsequently LEfSe was used to compare differential predicted metabolic functions between the classes.

### *In vitro* cultivation and genotyping of *Blastocystis* sp.

About 50 mg of fecal samples were inoculated into 3 ml Jones medium supplemented only with 10% horse serum as reported previously [22]. There were no antibiotics added to the medium to maintain the original bacterial composition. The cultures were incubated at 37˚C and screened daily for 5 to 7 days. The presence of *Blastocystis* sp. vacuolar forms was regarded as a positive sample. The xenic parasite cell culture was maintained *in vitro* and passaged every 3 to 4 days. Basic aseptic techniques were maintained throughout the culture process.

DNA was extracted from *Blastocystis* sp. *in vitro* cultures using Macherey Nagel Soil DNA extraction kit. Extracted DNA was used as a template to amplify and sequence the 18S small subunit ribosomal RNA gene (18S SSU-rDNA) at the length of 600bp using the protocols and primers described previously [23]. Five isolates of *Blastocystis* sp. were selected randomly to represent the symptomatic and asymptomatic groups for subsequent analysis.

### Alteration of microbiota surrounding *Blastocystis* sp.

Cells from three-day-old *Blastocystis* sp. (isolated from asymptomatic individuals) culture grown in 3 ml of Jones' medium supplemented with only 10% horse serum were spun at 1000 rpm to sediment *Blastocystis* sp. cells. No antibiotics were added to the Jones' medium. The supernatant containing mostly bacteria was isolated and washed with distilled water three times to lyse any remaining *Blastocystis* sp. cells. This bacterial suspension was then centrifuged at high speed and the pellet was re-suspended in 100 μl of Jones' medium with 10% horse serum.

Three-day-old *Blastocystis* sp. cells from symptomatic individuals were washed three times in PBS and counted to a concentration of $1 \times 10^5$ cells/ml in a final volume of 900 μl of Jone's medium. The bacterial suspension extracted earlier from asymptomatic *Blastocystis* sp. cultures were added and allowed to incubate at 37˚C. These steps were repeated every 3–4 days for 5 times before assessing the growth characteristics. The cells were then harvested and subjected to downstream processing. The experiment was repeated with asymptomatic *Blastocystis* sp. cell co-cultured with bacteria from symptomatic cultures. Four replicates were used in this experiment. Throughout this experiment, the xenic cultures that received bacterial suspension were regarded as co-cultured. Control experiments were those xenic cultures that were added with only sterilized Jones medium replacing bacterial suspension.

### Phenotypic characterizations

**Growth characterization.** Three-days old cells from co-cultured and control experiments were counted using heamocytometer chamber and inoculated into 1 ml medium with a final concentration of $1 \times 10^5$ cells/ml. The cells were counted every day for 10 days using trypan blue exclusion test to determine number of viable cells. The number of granular formation and amoebic formation per ml were counted.

**Colorimetric protease quantification assay.** Parasite isolates from co-cultured and control experiments were subjected to purification in order to recover parasites cells with minimal bacterial contamination. Purification was done through density-gradient centrifugation as described previously but with slight modification on the centrifugation speed [24]. The protease activity of the minimalized bacterial contamination was ensured to be at a negligible level as noted in the previous study [24]. The solubilized antigen of the purified *Blastocystis* sp. was extracted using the freeze-thaw technique. The purity and concentration were determined using Bradford Protein Assay (BioRad). The concentration of each sample was standardized to 0.1 mg/ml using a filter sterilized Jones medium before the assay. The specific protease activity was determined using azocasein colorimetric assay as reported previously [25,26].

**Colon cell proliferation analysis.** The antigen extracted previously were used to study colon cells proliferation. HCT 116 colonic cancer cells were obtained from American Type Cell Culture (ATCC) and maintained in RPMI medium supplemented with L-Glutamine, antibiotics and 10% fetal bovine serum (FBS). Colon cancer cells were maintained in T-25 vented culture flask at 37°C, 5% $CO_2$ and passaged every 4 to 5 days. To assess the proliferation, cells were standardized and seeded to 1000 cells/well in 100 µl using a 96-well plate as described by previous studies [27,28]. The cells were allowed to incubate at 37°C, 5% $CO_2$ for 24 hours. The antigens extracted earlier were added to the cells and incubated for another 48 hours. Cell proliferation was then determined using the MTT assay as described by the study cited above.

## Statistical analysis

The difference between groups was evaluated by comparing the mean using statistical tests. A student's t-test was conducted to compare the difference in alpha diversity and richness between groups using values derived from the diversity index. The student's t-test was also used to compare means of protease levels and cell proliferation. A p-value less than 0.05 is considered significant.

## Results

### Microbial profile analysis

A mean of 57828 sequences per sample from 50 individuals who were symptomatic (22/50) and asymptomatic(28/50) was obtained. The sequences were deposited in the National Library of Medicine (NCBI) as a BioProject with accession number PRJNA881789. A total of 1725 unique features were identified. The core phyla across the samples were Firmicutes and Bacteroidetes with more than 50% of relative abundance followed by Actinobacteria and Proteobacteria. At the genus level, the core genera observed across all samples were *Prevotella*, *Feacalibacterium*, *Bifidobacterium*, *Bacteroides*, and *Dialister* (S1 and S2 Figs).

### Variation of microbiota in symptomatic and asymptomatic individuals and effect of *Blastocystis* sp. colonization

Using the Shannon diversity index, significantly higher species diversity was observed in asymptomatic individuals (P = 0.028). The alpha rarefaction curve (at 5000 reads/sample) indicated that greater number of features seen in samples of asymptomatic individuals. Coverage index indicates the number of most abundant species occupying 50% of the community ecosystem. In this study, asymptomatic individuals showed significantly greater coverage compared to symptomatic individuals (P = 0.038). The evenness of the samples was measured using Pielou's evenness index. There was lower evenness seen in symptomatic individuals (P = 0.017). Beta diversity analysis through ordination by Non-metric multidimensional scaling (NMDS) using Canberra distance matrix showed clustering of symptomatic and asymptomatic samples (Stress = 0.17). The difference in diversity between samples was significant using PERMANOVA (P = 0.026) (Fig 1).

When the samples were generally classified into *Blastocystis* sp. infection status, we saw a significantly lower abundance of species in *Blastocystis* sp.-infected individuals using Chao1 richness estimator (P = 0.032) (Fig 2A). When the samples were grouped according to symptom, a significant difference in the abundance of species was only seen in asymptomatic individuals (P = 0.00026) while in the symptomatic group, there were no significant alterations to the species richness due to *Blastocystis* sp. infection (Fig 2B). Plotting of

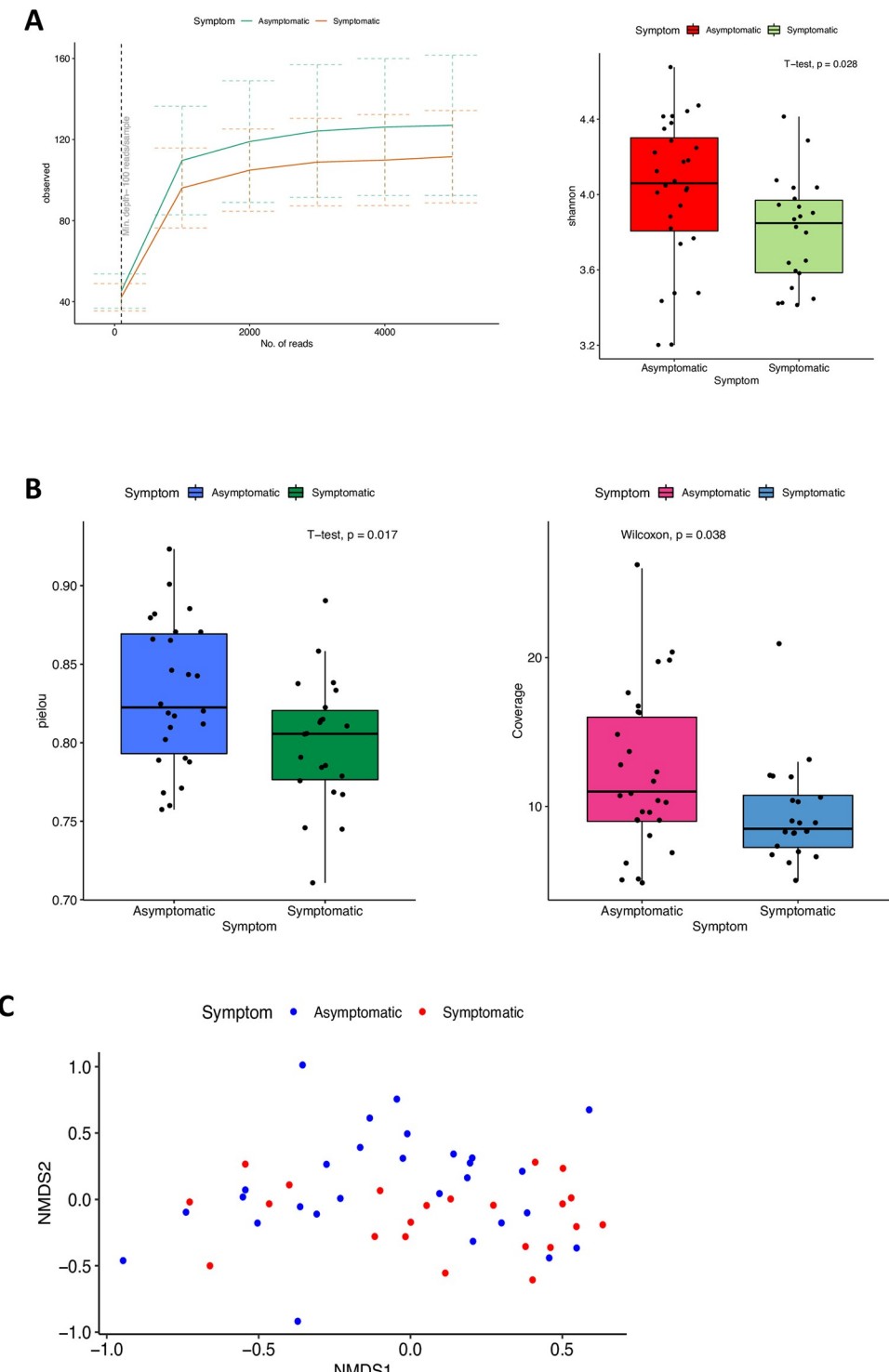

**Fig 1. Analysis of alpha diversity and beta diversity of gut flora isolated from symptomatic and asymptomatic individuals. A** Shannon diversity index showing higher diversity within gut flora of asymptomatic individual. **B** Pielou evenness and Coverage between the gut flora of symptomatic and asymptomatic individual. **C** Non-metric multidimensional scaling of gut microbial profile from symptomatic and asymptomatic individual ordinated based on Canberra distance matrix.

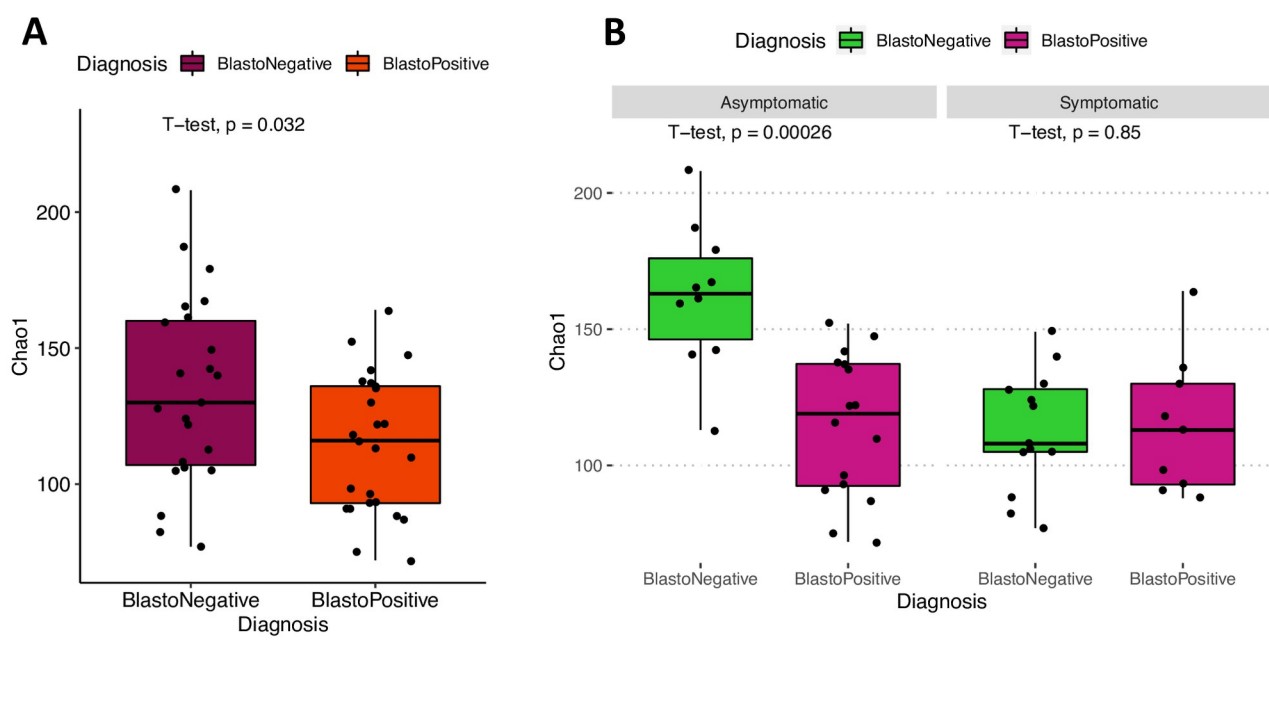

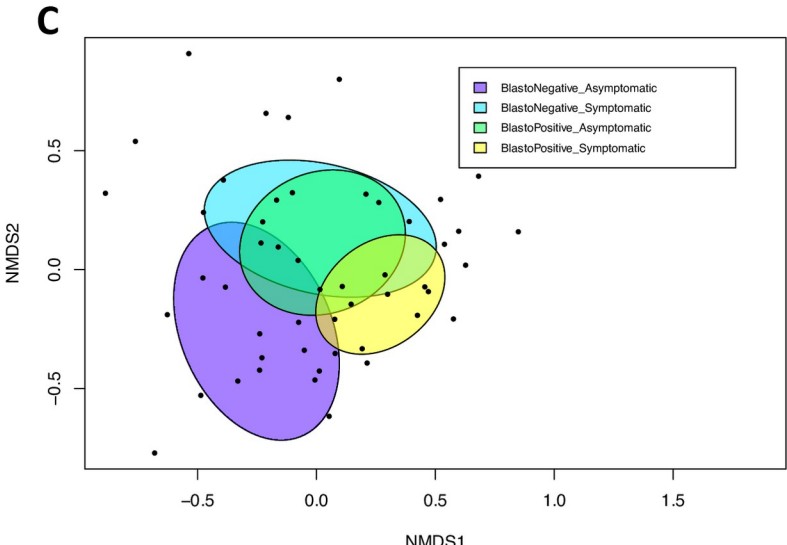

**Fig 2. Analysis of gut microbial diversity in *Blastocystis* sp. infection A** Chao1 diversity index showing higher diversity within gut flora of *Blastocystis* sp.-negative individual. **B** Chao1 comparing diversity in *Blastocystis* sp. infection in symptomatic and asymptomatic conditions. **C** NMDS plot including 95% confidence interval ellipses explaining 10% variation between groups (ADONIS: P<0.05, $R^2$ = 0.01). Each point represents samples ordained with the incorporation of abundance data of taxa up to species level using the Canberra distance matrix. The sample size of the groups is as following: BlastoNegative_Asymptomatic = 12;BlastoNegative_Symptomatic = 13; BlastoPositive_Asymptomatic = 16; BlastoPositive_Symptomatic = 9.

distance matrix using Canberra distance with the incorporation of abundance value revealed that infection of *Blastocystis* sp. regardless of symptoms has significant alterations in the gut microbiota (Fig 2C).

## Differential association of bacterial taxa to *Blastocystis* sp. infection in symptomatic and asymptomatic individual

LEfSe was deployed to determine the bacterial taxa that was differentially present in symptomatic and asymptomatic conditions and in *Blastocystis* sp.-positive and *Blastocystis* sp.-negative subjects. In this study, logarithmic LDA score of 3.0 was used as cut-off for the detection of important taxonomic differences. We found that *Prevotella* sp. was differentially abundant in symptomatic subjects while in asymptomatic group, bacterial phyla belonging to Firmicutes, Bacteroidotes, Verrucomicrobiota and Desulfobacteriota was abundant with LDA score beyond the fixed cut-off value (Fig 3A).

Within the symptomatic group, subjects that were *Blastocystis* sp.-positive showed that the family *Prevotellaceae* and *Ruminococcceae* were abundant whereas *Blastocystis* sp.-negative subjects showed an abundance of *Akkermansia* sp. and *Bacteroides* sp. (Figs 3B and S3). However, in the asymptomatic group, taxa from the phylum Firmicutes, specifically *Megasphaera* sp. and *Butyricicoccaceae* were differentially abundant in *Blastocystis* sp.-positive subjects while in *Blastocystis* sp. negative subjects, the increased abundance was in taxa belonging to the phyla Verrucomicrobiota, Firmicutes, and Bacteroidota (Fig 3C). The findings implicate that *Blastocystis* sp. in symptomatic and asymptomatic infection could be associated with different bacterial taxa.

## *Blastocystis* sp. colonization, and alteration in gut microbial functions

Microbial functions of the microbiota in the subjects were determined by using the inferred metagenomics obtained from PICRUSt. LEfSe was used to identify a differently abundant pathway in the samples. Microbial function in symptomatic individuals with *Blastocystis* sp. colonization generally showed an abundance of pathways involved in translation, nucleotide metabolism, metabolism of cofactors and vitamins, digestive systems, and also pathways involved in metabolic diseases compared to the microbiota without *Blastocystis* sp. which had pathways involved in transcription, signal transduction and lipid metabolism (Fig 4A). However, microbiota colonized by *Blastocystis* sp. in asymptomatic individuals had functional pathways abundant in metabolism of cofactors, vitamins and amino acids (Fig 4B). In general, pathways involved in replication and repair, nucleotide metabolism, translation, metabolic diseases and digestive system are found in *Blastocystis* sp.-positive subjects. The findings demonstrate that, it was not just the microbes that were differently abundant but also the metabolic functions that seemed to be different in *Blastocystis* sp.-colonized microbiota isolated from symptomatic and asymptomatic individuals.

## Genotyping and phenotypic characteristics of *Blastocystis* sp.

As reported in past studies [29], here we observed that *Blastocystis* sp. was found colonizing both symptomatic and asymptomatic individuals. All *Blastocystis* sp. isolated from this study belong to ST 3. The analysis of 18S partial length rDNA of *Blastocystis* sp. sequence analysis suggests that the ST 3 isolates belonged to allele 34. The 18S partial rRNA sequences of *Blastocystis* sp. ST3 isolated from symptomatic and asymptomatic individuals suggest close genotypic similarity and this means that any difference seen phenotypically would be solely due to external pressures. In this study, we assessed phenotypic expressions of the parasite from symptomatic and asymptomatic conditions. The phenotype was studied in terms of *in vitro* growth profile, specific protease activity, and ability to proliferate cancer cells. High peak cell count was observed specifically in *Blastocystis* sp. isolated from asymptomatic individuals than parasites isolated from the symptomatic individual (Fig 5A and 5C). Significantly greater total protease activity was seen in *Blastocystis* sp. isolated from symptomatic individuals. This was noticed in

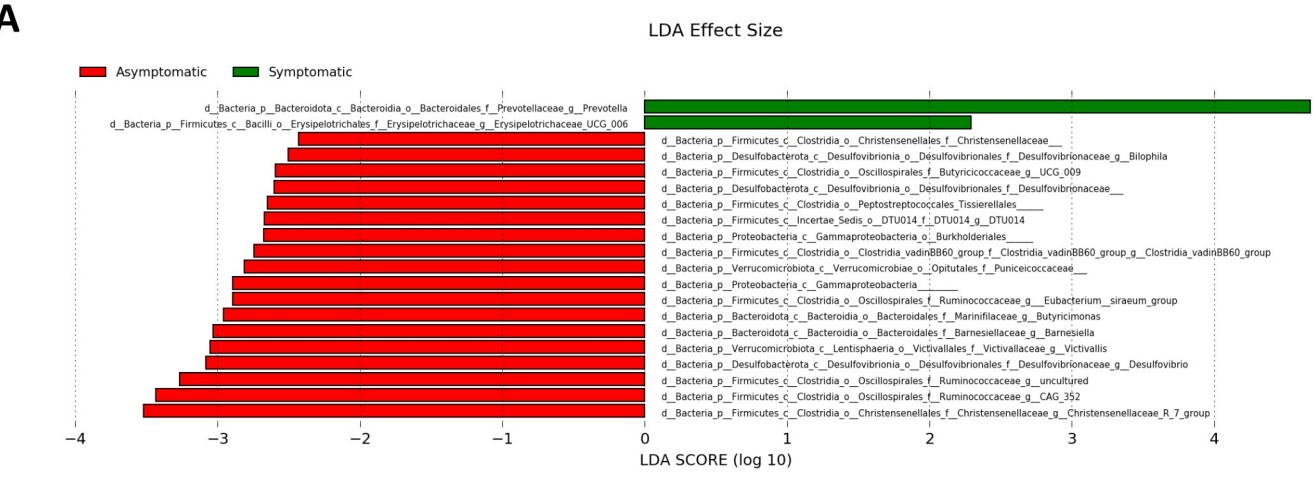

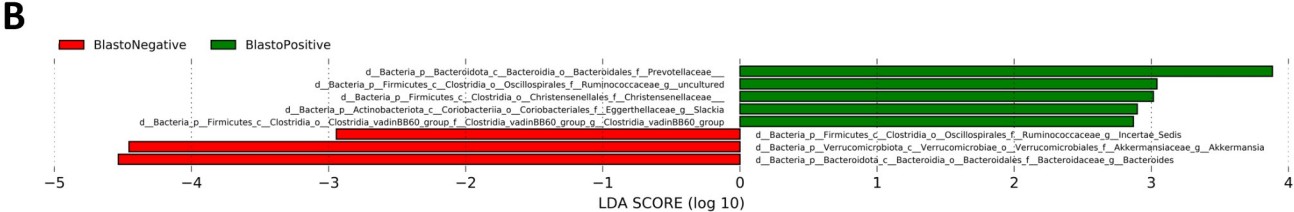

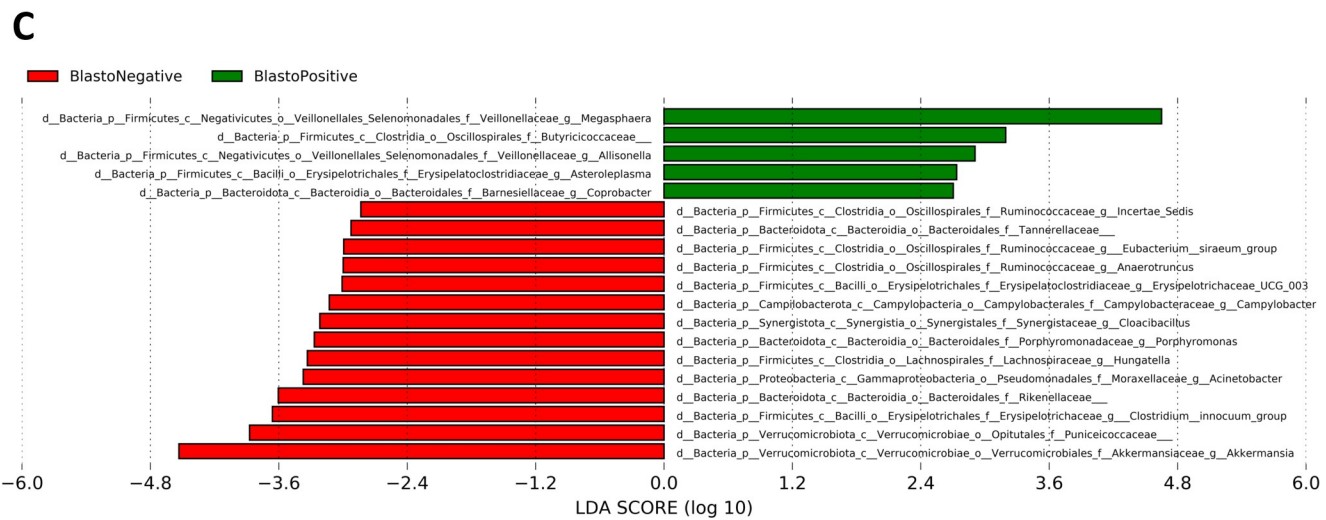

**Fig 3.** Differentially abundant bacteria taxa in **A** Gut microbiota of symptomatic and asymptomatic individuals. **B** Gut microbiota of symptomatic individuals with and without *Blastocystis* sp. **C** Gut microbiota of asymptomatic individuals with and without *Blastocystis* sp.

the control experiments in Fig 6. We also observed that the *Blastocystis* sp. obtained from symptomatic individuals had a predominance of cysteine protease whereas the parasite cells isolated from asymptomatic individuals possessed serine protease predominantly (Fig 6A).

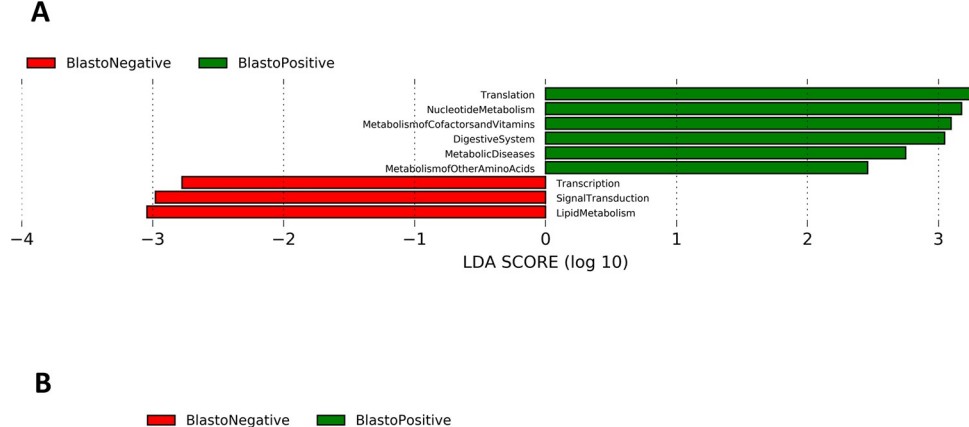

**Fig 4.** Influence of *Blastocystis* sp. on the gut microbial function in **A** symptomatic and **B** asymptomatic individuals.

## Influence of bacteria on *Blastocystis* sp.

*Blastocystis* sp. from symptomatic individuals co-cultured with bacterial suspension from asymptomatic individuals showed increased growth of parasite numbers compared to the parasite isolates without introducing bacterial suspension from asymptomatic isolates. The average peak cell count of 2.46 x 10$^6$ cells/ml increased about 3-fold to 6.54 x 10$^6$ cells/ml. Whereas *Blastocystis* sp.

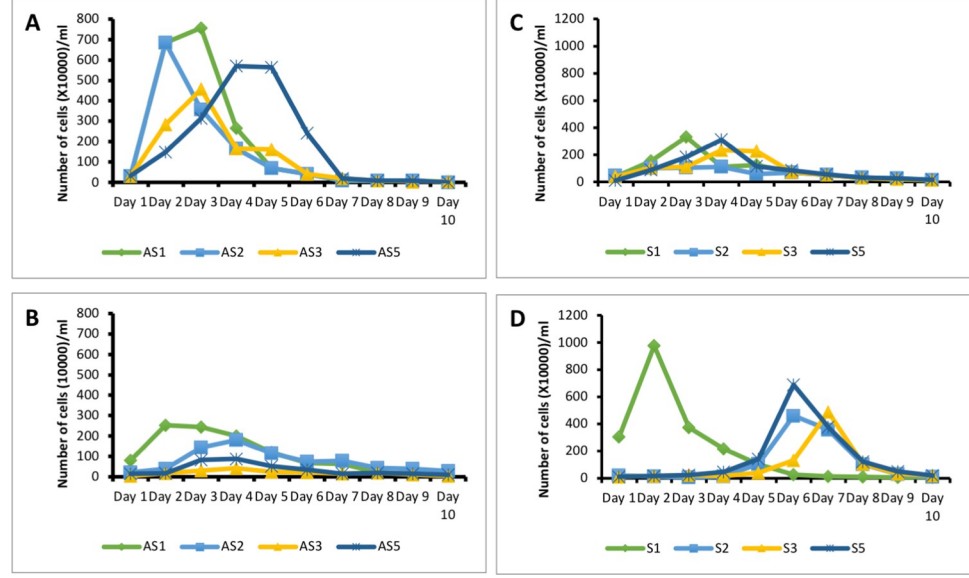

**Fig 5. Growth profile of *Blastocystis* sp. upon introduction of bacterial suspension from symptomatic and asymptomatic culture. A** Growth profile of parasites obtained from asymptomatic individuals. **B** Growth profile of parasites obtained from asymptomatic individuals co-cultured with bacterial suspension of symptomatic parasite culture. **C** Growth profile of parasites obtained from symptomatic individuals. **D** Growth profile of parasites obtained from symptomatic individuals co-cultured with bacterial suspension of asymptomatic parasite culture.

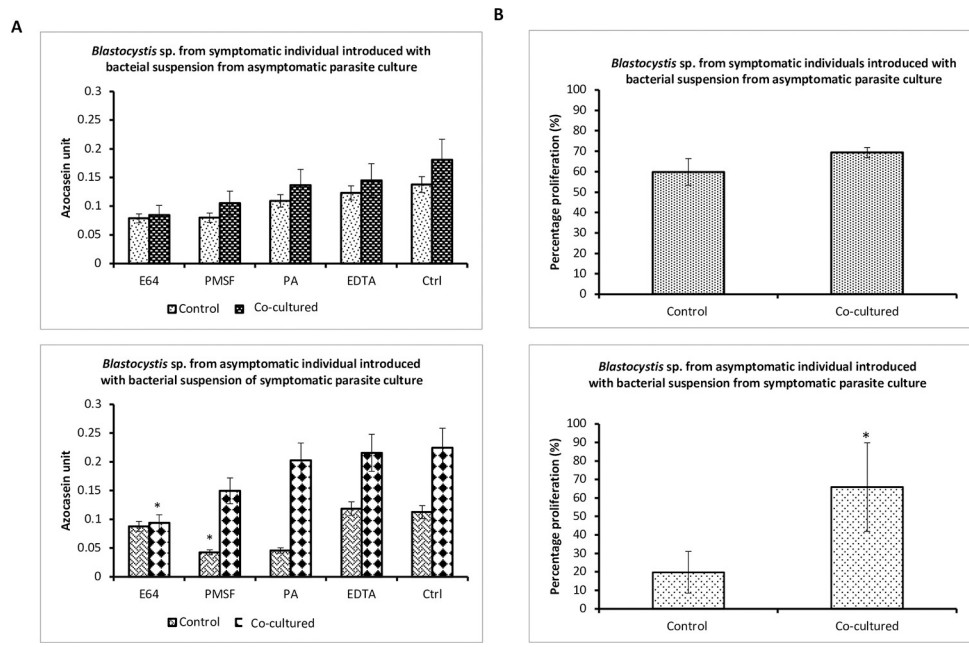

**Fig 6. Influence of bacterial alteration resulting in variation of protein expression. A** Changes in specific protease activity in *Blastocystis* sp. isolated from symptomatic and asymptomatic individuals after the introduction of bacterial suspension from asymptomatic and symptomatic parasite culture. Values are expressed as mean±SD from 4 replicates. *P<0.05 in Student's t-test for comparison with control. Note: E64 = cysteine protease inhibitor; PMSF = serine protease inhibitor; PA (Pepstatin A) = aspartic protease inhibitor; EDTA = metalloprotease inhibitors. Ctrl: Protease activity without addition of inhibitors. **B** Cell proliferation by solubilized antigen of *Blastocystis* sp. isolated from symptomatic and asymptomatic individuals after the introduction of bacterial suspension from asymptomatic and symptomatic parasite cultures. Values are expressed as mean±SD from 3 replicates. *P<0.05 in Student's t-test for comparison with control. Co-cultured: Experiments with introduction of bacterial suspension. Control: Experiments with introduction of sterile Jones medium instead of bacterial suspension.

obtained from asymptomatic individuals, which had high cell numbers showed a reduction in cell count upon introducing bacterial suspension from symptomatic individual. The average peak cell count decreased 4-folds from 6.17 x 10$^6$ cells/ml to 1.45 x 10$^6$ cells/ml (Fig 5).

Protease activity in *Blastocystis* sp. isolated from symptomatic individuals co-cultured with bacterial suspension extracted from asymptomatic individual showed only a slight increase, which was insignificant. However, isolate obtained from asymptomatic individuals when cultured with bacterial suspension isolated from symptomatic individuals showed significant increase in the protease activity (from 0.085 to 0.2789). The increase was statistically significant using Student's t-test when compared to the control (P = 0.029). *Blastocystis* sp. isolated from asymptomatic individuals initially possessed significant predominance of serine protease. When bacterial suspension from symptomatic individuals was introduced to *Blastocystis* sp. obtained from asymptomatic individual there was an increase in the cysteine protease. This increase was found to be significant using the Student's t-test (P<0.05) (Fig 6A).

Regarding the ability of *Blastocystis* sp. antigens to promote colonic cell proliferation, antigens isolated from symptomatic individuals that were cultured with bacteria from asymptomatic *Blastocystis* sp. culture showed insignificant proliferation compared to the control. Nonetheless, antigens from *Blastocystis* sp. isolated from asymptomatic individuals that were co-cultured with bacteria from symptomatic individuals produced significantly greater colonic cell proliferation than antigens from *Blastocysits* sp. isolated from asymptomatic individuals in autochthonous culture. There was about 3-fold increase from 19.8% proliferation to 65.8% (Fig 6B).

## Discussion

*Blastocystis* sp. has been reported to have an intricate relationship with its surrounding bacteria [30]. A previous study had orally inoculated axenic, monoxenic and xenic *Blastocystis* sp. from symptomatic individuals into germ-free guinea pigs. It was found that about half of the rats inoculated with xenic parasite developed infections with watery diarrhea for more than a week duration and increased cellularity at the lamina propria region. In contrast, the rats inoculated with monoxenic had an infection and none was infected in rats inoculated with axenic *Blastocystis* sp.[31]. This finding was one of the earliest to highlight the importance of accompanying gut bacteria in *Blastocystis* sp. infection.

In this study, the phyla Firmicutes and Bacteroidetes were most predominant among all the subjects. This is consistent with a previous study on a similar Malaysian population [32,33]. In general, regardless of *Blastocystis* sp. infection, we found a significant difference in alpha and beta diversity, confirming that bacterial composition in symptomatic and asymptomatic samples is distinct. This confirms that gastrointestinal symptoms are associated with low species richness. LEfSe and relative abundance analysis further confirms alteration in Firmicutes/Bacteroidetes (F/B) ratio. Decreased F/B ratio seen in symptomatic individuals suggests dysbiosis, commonly also seen in inflammatory bowel disease (IBD) patients [34].

Our findings demonstrated that *Blastocystis* sp.-infected individuals, regardless of symptoms, had decreased alpha diversity and Pielou's evenness. As observed in our recent study, greater amoebic forms and surface fuzzy coat commonly seen in symptomatic isolates [35] suggest a greater interaction with bacteria in these isolates which could contribute to the alteration of microbiota. On the other hand, lower peak cell numbers in the growth profile of symptomatic isolates implicate potential inhibition from accompanying microbiota. These observations suggest a bidirectional interaction between *Blastocystis* sp. and gut microbiota. However, more data on *Blastocystis* sp.-gut microbiota across multiple populations is required to corroborate this interaction. A similar study done on pooled symptomatic and asymptomatic populations showed contrasting results where no difference in alpha diversity was detected [36]. This discrepancy could have been potentially contributed by differences in environment and the studied population. Since subtype-influenced associations to gut microbiota have been demonstrated by Tito et al. [7] it is highly likely that the discrepancy is due to analyses being carried out on multiple *Blastocystis* sp. subtypes (ST 1–7). In this study, the association seen is unique and specific to only *Blastocystis* sp. ST3. While most other studies have compared *Blastocystis* sp.-gut microbiota association in diseased or healthy group [5,6,37,38], our study for the first-time reported association of a single subtype (ST 3) of *Blastocystis* sp. to symptomatic and healthy individuals.

A study by Nagel et al. on *Blastocystis* sp. from irritable bowel syndrome patients revealed insignificant influence on the gut microbiota [38]. Similarly, in this study, the presence or absence of *Blastocystis* sp. in symptomatic group did not significantly influence bacterial diversity but changed the abundance of certain bacterial taxa suggesting alterations in bacterial composition. However, in asymptomatic individuals, we saw a significant alteration in gut microbial diversity and composition in *Blastocystis* sp. infection. Interestingly, different composition of bacteria was seen to be associated with *Blastocystis* sp. in symptomatic and asymptomatic infections. In symptomatic individuals, bacteria from the family of Prevotellaceae and Rumunicoccaceae were predominant in *Blastocystis* sp. colonization. Our study is similar to previous reports where Prevotellaceae were positively associated with *Blastocystis* sp. colonization [5]. Studies have associated bacteria from Prevotellaceae with inflammatory disorders [39] and symptomatic *Entamoeba histolytica* infection [40]. However, its role especially in symptomatic *Blastocystis* sp. infection needs further exploration as the parasite often presents

features such as amoebic forms and a sticky surface coat [41,42] implicating enhanced interaction with bacteria.

Asymptomatic individuals with *Blastocystis* sp. colonization were associated with a predominance of bacteria belonging to mainly Firmicutes with reduced diversity. A recent study on the Iranian population has reported similar findings where harmful bacteria were elevated in asymptomatic *Blastocystis* sp. infection [43]. The findings by Nieves-Ramirez et al [6] showed increased diversity in asymptomatic *Blastocystis* sp. infection, although there was a similar increase in Firmicutes. Population heterogeneity could be a reason as the study was conducted in the Mexican rural population while the current study was done on the urban population in Malaysia. A significant difference in bacterial composition between Malaysian and western populations [33] suggests the contrasting findings between this study and other studies on gut microbiota-*Blastocystis* sp. association [5,37]. However, the reduction in richness in asymptomatic *Blastocystis* sp. infection could be best explained ecologically by an alternative stable state [44], whereby perturbation in gut microflora results in the establishment of a different stable state with associated dynamics such as population fluctuations. This stable state may contribute to specific immunological adjustments as previous reports have noticed reduced fecal calprotectin, IgA level [6], and neutrophil levels [45] in asymptomatic *Blastocystis* sp. infection. Whether *Blastocystis* sp. instigates an anti-inflammatory environment for persistent asymptomatic colonization by altering gut bacterial composition warrants more study.

PICRUSt algorithm is commonly used for the functional prediction of the intestinal microbiota [33,46]. In adjunct to microbial diversity and composition, we used the PICRUSt algorithm to further add dimension to metabolic functions in gut microbiota and its alteration after *Blastocystis* sp. infection in studied subjects. The outcome, for the first time, implies that *Blastocystis* sp. could be related to modifications of resulting microbial functional pathways. This is likely due to the alteration of microbial composition in *Blastocystis* sp. infection. We postulated that asymptomatic *Blastocystis* sp. infection could be associated with an alternative stable state. In symptomatic individuals, *Blastocystis* sp. altered microbial composition despite the diversity not being significantly affected. Studies suggested that altered bacterial composition can influence how metabolites are processed, resulting in the metabolic pathway and profile changes [2]. Evidence from this study suggests the same, as significant change in metabolic processes are observed in *Blastocystis* sp. infection. However, the predicted microbial functional pathway only offers preliminary access to understanding microbiota function. Greater depth and details provided by the metabolomic approach and whole genome sequencing would be essential in identifying genes involved in the specific metabolic pathway and metabolite interactions during *Blastocystis* sp. infection.

To date, no studies have reported the influence of accompanying bacteria on *Blastocystis* sp. cells. Several studies have suggested important roles of accompanying bacteria in the pathogenesis of intestinal protozoan parasite [9,12]. These studies, however, do not mimic the natural condition of the gut as the parasite cells used were axenic. Hence, the changes seen in the parasites may not translate to the real-time scenario. Here we report the effects of altering the autochthonous bacterial composition in *Blastocystis* sp. culture. We found that parasite cells isolated from asymptomatic individuals resembled the cells from symptomatic ones upon consistent introduction of bacterial suspension from symptomatic individual and vice versa. These findings demonstrate the role of bacteria in influencing *Blastocystis* sp. up to protein expression levels where the solubilized protease levels and ability of antigens to proliferate colon cancer cells *in vitro* were also altered. Although the role of proteases is inconclusive in *Blastocystis* sp. infection, evidence of degradation of secretory immunoglobulin A [47] and activation of IL-8 gene expression [48] and well-studied pathogenic roles in other intestinal parasites [49,50] potentially implicate it as a virulent factor. Evidence on bacteria engulfing

amoebic forms and increased protease activity [24] in *Blastocystis* sp. as well as lipopolysaccharide (LPS)-induced toll-like receptor activation [51] supports the obligatory role of bacteria in pathogenic characteristics in *Blastocystis* sp.

Earlier studies demonstrated the presence of physical features such as sticky surfaces, fuzzy coats, and amoebic morphologies indicating interactions with bacteria [52]. With current findings, we are certain that this interaction contributes to shaping the phenotypic feature of *Blastocystis* sp. cells. Several studies have used phenotypic features to ascribe pathogenic potentials in *Blastocystis* sp. ST3 when isolated from diseased and healthy groups [41,53,54]. The resulting phenotype reflects a specific microbiota composition, either due to various host-related factors or possibly influenced by *Blastocystis* sp. itself. The latter is highly likely, as we have seen here and in other recent findings, that *Blastocystis* sp. modifies the bacterial composition [6,55], although its influence on diversity is contradictory. Even so, nothing is conclusive until the mechanism of specific phyla in cross-talk with *Blastocystis* sp. is elucidated. Studies on the pathogenicity of *Blastocystis* sp. have been contradicting for a long time. However, case studies [56] reporting improvement of symptoms upon the extermination of this organism via drug treatment suggest a pathogenic role that could be restricted to only some individuals or certain gut microbial environments. Studies thus far have reported that pathogenic potentials are being assessed in terms of variations in the growth profile, cysteine protease activity, and ability to proliferate cancer cells [57]. Our finding showed that these factors, disparate in *Blastocystis* sp. isolated from symptomatic and asymptomatic individuals, are largely dependent and altered by the parasite's microbial surroundings. We propose that the pathogenic characteristics may not be wholly exerted by the parasite itself but influenced by factors such as the gut microbiota as well.

Increasing number of studies is beginning to show differences in gut microbiota due to various factors [58]. An individual may undergo alterations in gut microbial environment as a consequence of changing dietary intakes, life stresses, medications, travel and migrations, which are rampant in recent years. A diverse and balanced microbiota profile provide protection to the mucosa [59] and secrete metabolic products such as short chain fatty acids (SCFA) that promote health [60]. Parasitic cells colonizing in such environment may remain asymptomatic [59]. When there is a change in the environment, especially when triggered by certain diet, antibiotic consumption or stress, the microbiota may be altered [1]. This influences the colonizing organism initially harmless to be pathogenic (Fig 7). Although this postulation was derived from the correlation of data without a causal relationship, we believe this is the way forward in understanding the role of *Blastocystis* sp. in disease and health.

Our study is limited in terms of sample size. This is due to difficulty in obtaining a single subtype of *Blastocystis* sp. and maintaining the parasite cells *in vitro*. A similar experimental design, applied to larger sample size, yields more conclusive evidence. This study is also limited in terms of the use of molecular diagnostics for the detections of *Blastocystis* sp. colonization. Therefore, it is possible that *Blastocystis* sp.-free group may have individuals who were infected with this organism but was not captured by *in-vitro* cultivation technique. Also, the quantitative burden of *Blastocystis* sp. could not be compared between the groups. However, this study, for the first time, has demonstrated gut variation associated with a single *Blastocystis* sp. subtype. Our study has also shown for the first time the influence of autochthonous bacterial alteration on the phenotype of *Blastocystis* sp. ST3 cells. In the future, it is essential to characterize the bacterial taxa in close interaction with *Blastocystis* sp. ST3 and its role in symptomatic and asymptomatic infections.

## Conclusion

Many recent studies focused on the effect of *Blastocystis* sp. in altering the gut microbiota [5,61], however, this is the first study to demonstrate the influence of microbial environment

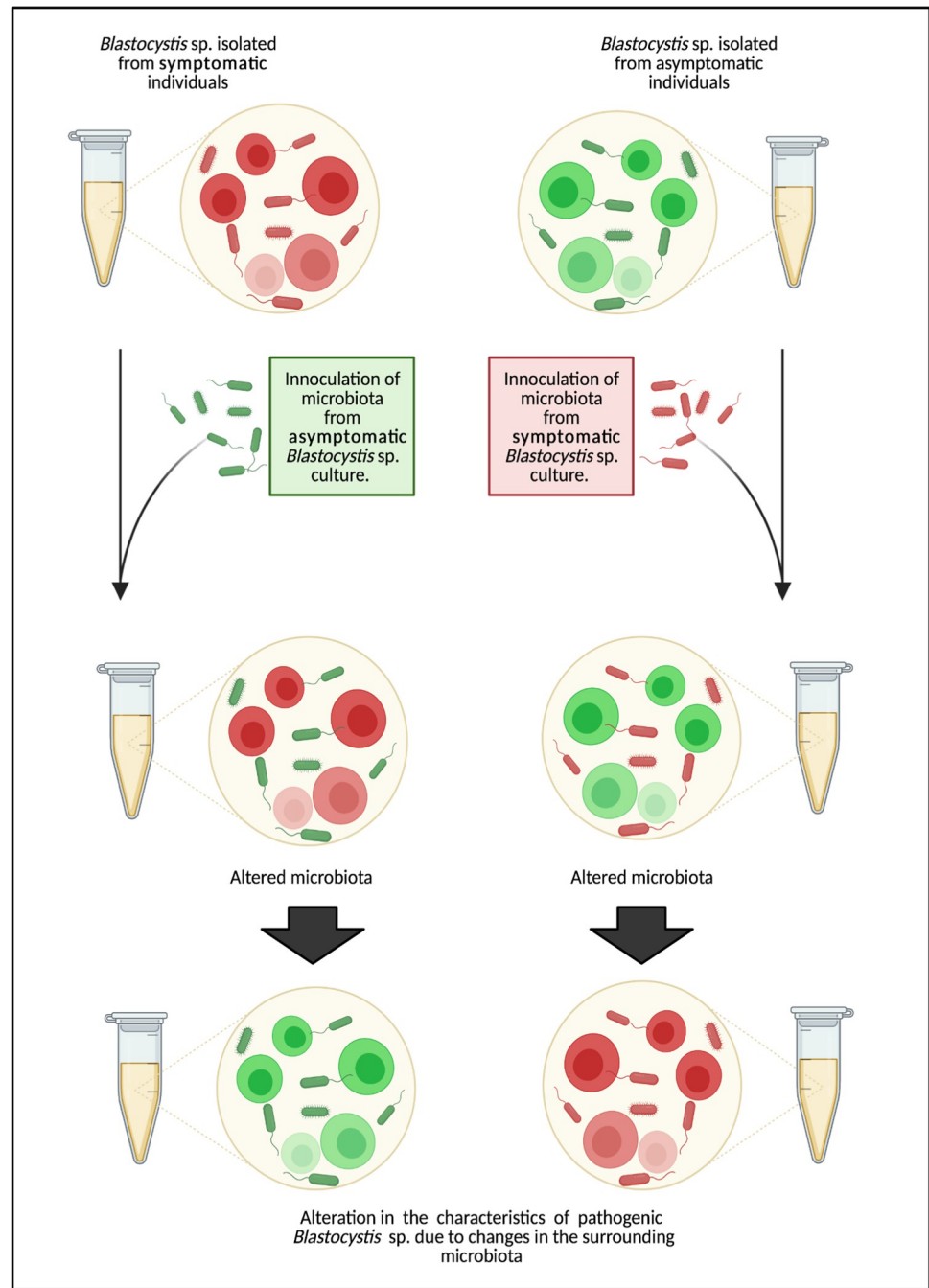

**Fig 7. Schematic diagram demonstrating the effect of microbiota in alternating the characteristic of *Blastocystis* sp.**

on this prevalent intestinal protozoon. The findings open new vistas in understanding parasite-bacteria interaction, which could help us understand better the pathogenicity of *Blastocystis* sp. We postulate that the interactions seen between specific intestinal microbiota and *Blastocystis* sp. influence whether the protozoa will function in a commensal or parasitic role. This study also provides preliminary evidence of a typical intestinal protozoan reverting from a harmless organism to a harmful one.

## Supporting information

**S1 Table. The demographic profile of participants (n = 50) recruited into the study.**
(XLSX)

**S1 Fig.** Breakdown of core microbiota at (**A**) phylum and (**B**) genus level in symptomatic and asymptomatic individuals.
(TIFF)

**S2 Fig.** Breakdown of rare microbiota at (**A**) phylum and (**B**) genus level in symptomatic and asymptomatic individuals.
(TIFF)

**S3 Fig.** Breakdown of core microbiota at (**A**) phylum and (**B**) genus level in symptomatic and asymptomatic individuals with and without *Blastocystis* sp. colonization.
(TIFF)

## Acknowledgments

We would like to thank the staff of Pantai Medical Hospital Specialist Clinic and Department of Parasitology, Faculty of Medicine, University Malaya.

## Author Contributions

**Conceptualization:** Arutchelvan Rajamanikam, Chandramathi Samudi, Suresh Kumar Govind.

**Data curation:** Arutchelvan Rajamanikam, Chandramathi Samudi, Suresh Kumar Govind.

**Formal analysis:** Arutchelvan Rajamanikam, Mohd Noor Mat Isa, Chandramathi Samudi, Sridevi Devaraj.

**Funding acquisition:** Suresh Kumar Govind.

**Methodology:** Arutchelvan Rajamanikam, Sridevi Devaraj.

**Project administration:** Arutchelvan Rajamanikam.

**Validation:** Arutchelvan Rajamanikam.

**Writing – original draft:** Arutchelvan Rajamanikam.

**Writing – review & editing:** Arutchelvan Rajamanikam.

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
