## [Decision Letter · Decision Letter 0]

10 Aug 2022

Dear Dr Rajamanikam,

Thank you very much for submitting your manuscript "A harmless organism in the gut can be triggered to be harmful" for consideration at PLOS Neglected Tropical Diseases. As with all papers reviewed by the journal, your manuscript was reviewed by members of the editorial board and by several independent reviewers. In light of the reviews (below this email), we would like to invite the resubmission of a significantly-revised version that takes into account the reviewers' comments. 

Apologies for delays in returning this manuscript. Two reviewers have carefully reviewed your manuscript, and both have determined that your manuscript could be accepted with major revisions. Some clarifications in the text and aesthetic modifications of the figures are required, but some methodological weaknesses need to be acknowledged as well. Finally please ensure that sequence data is deposited on a public repository and accession numbers are included in the text.

We cannot make any decision about publication until we have seen the revised manuscript and your response to the reviewers' comments. Your revised manuscript is also likely to be sent to reviewers for further evaluation.

Sincerely,

Matthew Brian Rogers, Ph.D.

Academic Editor

Shan Lv

Section Editor

Apologies for delays in returning this manuscript. Two reviewers have carefully reviewed your manuscript, and both have determined that your manuscript could be accepted with major revisions. Some clarifications in the text and aesthetic modifications of the figures are required, but some methodological weaknesses need to be acknowledged as well. Finally please ensure that sequence data is deposited on a public repository and accession numbers are included in the text.

Reviewer's Responses to Questions

**Key Review Criteria Required for Acceptance?**

**Methods**

-Are the objectives of the study clearly articulated with a clear testable hypothesis stated?

-Is the study design appropriate to address the stated objectives?

-Is the population clearly described and appropriate for the hypothesis being tested?

-Is the sample size sufficient to ensure adequate power to address the hypothesis being tested?

-Were correct statistical analysis used to support conclusions?

-Are there concerns about ethical or regulatory requirements being met?

Reviewer #1: -Are the objectives of the study clearly articulated with a clear testable hypothesis stated? Yes

-Is the study design appropriate to address the stated objectives? Yes

-Is the population clearly described and appropriate for the hypothesis being tested? No

A) please present the demographic characterizations of both groups in a table; B) please indicate whether healthy individual or infected groups had underlying diseases (e.g., diabetes, autoimmune disorders, etc.) or other gut infections; C) please indicate B. hominis group had single infection and did not have co-infection with other gut pathogens or parasites. 

-Is the sample size sufficient to ensure adequate power to address the hypothesis being tested? Yes

-Were correct statistical analysis used to support conclusions? Please add a statistical analysis section at the end of material and methods. 

-Are there concerns about ethical or regulatory requirements being met? No

Reviewer #2: - the authors need to articulate in the methods the definition what is meant by symptomatic and asymptomatic. Presumptively, this means intestinal symptoms (? diarrhea). Were these definitions made a prior or only after participant recruitment and questionnaire review. Given the potential range of 'subclinical' associations between a disease state and presence of Blastocystis described in the introduction it is critical that the report be specific about what disease condition is being studied. 

- it appears that there were 50 individuals and each provided one sample. Please indicate this clearly.

- 'maintaining utmost anonymity' is vague. Please elaborate on why written consent was not required and the procedures for de-identification of samples

- The recruitment procedures and any inclusion/exclusion criteria are absent. Coupled with a lack of definition for symptomatic vs asymptomatic, it makes it difficult to appraise the assignment of different groups.

- Please include a section for assignment of Blastocystis carriage status. What assignment based on culture recovery of parasites? qPCR data for Blastocystis would be more sensitive and allow for analyses based on parasite burden.

- Line 187 implies that there was a screen for pathogens in the stool samples, yet a list of what pathogens and by what method is not provided. If Blastocystis is a common co-pathogen, this should be considered in the interpretation of the 16S sequencing data. What is meant in line 188-189, eg. in Line 189, which groups?

-Please clarify if antibiotics are present in the Jones media. Though a reference for the media is provided, given the emphasis of this paper on parasite-bacteria interactions it is important to explicitly indicate what if any antibiotics were present in the media.

- Line 192. delete 'asymptomatic' as an adjective modifying Blastocystis, the parasite. This allows the authors to delete the paranethetical (ie. 3-days-old Blastocystis sp isolated from asymptomatic individuals...)

**Results**

-Does the analysis presented match the analysis plan?

-Are the results clearly and completely presented?

-Are the figures (Tables, Images) of sufficient quality for clarity?

Reviewer #1: -Does the analysis presented match the analysis plan? Yes

-Are the results clearly and completely presented? Yes

-Are the figures (Tables, Images) of sufficient quality for clarity? Yes

Please add accession numbers if you deposited your sequences in GenBank.

Reviewer #2: -Figures. It is customary to show the letter designation for figure panels in the upper left, not the upper right.

- Line 238. If separating the symptomatic from asymptomatic, groups, then report the mean of all 50 participants and separately the mean in symptomatic vs asymptomatic for each group. Likewise, a break down comparison of phyla and genera between symptomatic and asymptomatic should be performed.

- by 'features' do the authors mean 'assigned sequence variants'? What is a 'feature'?

-Figure 2: Please include a sentence about the 'n' of Blastocysts positive vs neg (A) and including by clinical status (B). C, What determined the circle sizes? Where are the individual samples? Is this absolute range of dots (where are the dots?), 95% CI?, some other measure of range?

-Lines 318-324: please reference to a figure or data to substantiate the claims made in the text. In addition, please comment on what methods were used to validate that the phenotypic differences between Blastocystis isolated from symptomatic versus asymptomatic participants were due to Blastocystis specifically and not the co-cultured bacteria in these xenic cultures. Can the authors axenize Blastocystis?

- Lines 327-334: does filtered or heat-killed bacterial suspension have a similar result?

- Line 336: please indicate how the authors know that the protease activity is a function of Blastocystis rather than the bacteria present in symptomatic xenic cultures. protease activity could be a composite of Blastocystis and bacteria/other microbes present in the culture.

- Line 348: How was the blastocysts antigen prepared and what measurements were done to confirm its purity compared with other proteins, including bacterial antigens, that may be present in the extract. Also, numerically, the difference in proliferation between Blastocystis antigen from asymptomatic (20%) and symptomatic (60%) stool is similar (and with less variability) to asymptomatic (20%) and co-culture with bacteria (~65%). Please clarify. Also, both figures in 6B read 'bacterial suspension from asymptomatic parasite culture'. This seems inconsistent with the text. Finally, is 'control' bacteria-free (axenic) or is 'control' with bacteria in the autochthonous xenic culture used to recover Blastocystis?

**Conclusions**

-Are the conclusions supported by the data presented?

-Are the limitations of analysis clearly described?

-Do the authors discuss how these data can be helpful to advance our understanding of the topic under study?

-Is public health relevance addressed?

Reviewer #1: -Are the conclusions supported by the data presented? Yes

-Are the limitations of analysis clearly described? No

Please say your strengths, limitations, and suggestions for future researchers as a concise paragraph before conclusion section. 

-Do the authors discuss how these data can be helpful to advance our understanding of the topic under study? somewhat

-Is public health relevance addressed? Yes

Reviewer #2: - The authors have performed multiple experiments to derive observations of potential Blastocystis - bacteria interactions. These questions are complex and the experiments are by extension challenging. 

- That Blastocystis is accompanied by decreased alpha diversity and Pielou's evenness does not distinguish causality from association. In addition to potential strain-dependent influences the authors reference in lines 380-385, there are potentially other factors (environmental, individual patient, population, and methodological) that could explain a discrepancy between findings in this study and other manuscripts. Another interpretation is that the lower diversity stool selected for assignment of Blastocystis status (and therefore a molecular method to assign Blastocystis status should be pursued). It is harder to culture Blastocystis ST3 from some stool (ie. symptomatic as shown in Figure 5) than other stool and perhaps not surprisingly stool with less microbial diversity is less likely to have bacteria that inhibit Blastocystis growth.

-The conclusion overstates what was observed. Line 506 extends beyond the data observed in this manuscript. There is no measure of immune barrier and immune response in this study and such a conclusion is only conjecture. Also, the designation of Blastocystis as either commensal or parasite based on the data presented is not possible. There is no evidence that a) Blastocystis is contributing to a disease condition in the participants or that b) the in vitro readouts are sufficient to designate a 'parasitic' phenotype. Rather than stating 'here we present', the authors could raise their new hypothesis that interactions between specific intestinal microbiota and Blastocystis influence whether the protozoan is functioning in a commensal or parasitic role. Finally, Line 510, I think the authors would agree that the evidence presented is not strong. The study may be suggestive of the statement, but the presented data is not a complete mechanism.

**Editorial and Data Presentation Modifications?**

Reviewer #1: (No Response)

Reviewer #2: (No Response)

**Summary and General Comments**

Reviewer #1: The study is interesting and well designed. However, I have some comments and suggestions:

1. Regarding stool sample collection from case and control groups: A) please present the demographic characterizations of both groups in a table; B) please indicate whether healthy individual or infected groups had underlying diseases (e.g., diabetes, autoimmune disorders, etc.) or other gut infections; C) please indicate B. hominis group had single infection and did not have co-infection with other gut pathogens or parasites. 

2. Please add a statistical analysis section at the end of material and methods.

3. Please add accession numbers if you deposited your sequences in GenBank. 

4. Please read and use the recent paper regarding asymptomatic B. hominis and gut bacteria composition: Behboud, et al. "Alteration of gut bacteria composition among individuals with asymptomatic Blastocystis infection: A case-control study." Microbial Pathogenesis (2022): 105639.

5. Please say your strengths, limitations, and suggestions for future researchers as a concise paragraph before conclusion section.

Reviewer #2: Overall the authors present an intriguing concept that microbiota-protozoa interactions are critical for understanding intestinal protozoan behavior (in this case Blastocystis) and disease potential. These are difficult hypotheses to test and prove, and the authors used an in vitro system of mixed stool culture derived Blastocystis with bacteria present in the same stool or from stool from participants with a different clinical status. The strength of the study is the attempt to disentangle these complex microbe-microbe interactions. Weaknesses, however, are several including: incomplete descriptions of the participant cohort (especially the definition of symptomatic or asymptomatic), apparent lack of a molecular assay to assign Blastocystis status (and therefore the designation of Blastocystis status is contingent upon culture viability which that authors demonstrate may be influenced by autochthonous bacteria in the stool), apparent lack of axenic Blastocystis controls, and lack of reporting what measurements or manipulations were performed to indicate that the phenotypic changes seen from different mixes of microbial communities were derived from changes in Blastocystis rather than derived from the other microbes in the xenic assay system. Additional experiments are therefore needed to reach the authors' final conclusions. 

Additional specific comments:

--Abstract: multiple changes are necessary 'microbiome' to 'intestinal microbiota'. Reserve the use of the term 'microbiome' in reference to descriptions of the intestinal microbial community genomic structure.

--"healthy" could easily be a misclassification of individual status and should be avoided. Rather these 'healthy' individuals are Blastocystis ST3 negative controls. 

--change line 44. This reads as an overstatement and 'symptomatic' or 'asymptomatic' refer to host syndromes, not characteristics of a parasite. It would be better to just state the observation without assigning presumed consequences on the host:eg. ".... demonstrated diminished growth and diminished expression of putative virulence genes"..

--line 47 'microbial diversity' appears to refer to the genomic (16S amplicon?) diversity. This should be more clearly stated.

Introduction

-The introductory paragraph is difficult to read, in part because of the sentence structure, and in part because of the complexity of the concepts. First, there are plenty of studies to indicate that plasticity in response to dietary and environmental influences is a fundamental characteristic of intestinal microbiota (and therefore not unexpected). The first sentence also inappropriately lumps direct influences with established influences on intestinal microbial community structure (ie. diet) with those that are indirect, or require several steps to prove causality (ie. climate change, to this reviewer's knowledge had not been proven to directly alter intestinal microbiota). It is important when introducing this concept that the authors take care to delineate what is known and verifiable from what is speculation (even if plausible). In Line 60 and line 62 it is also not clear what specific influences on the host are relevant for the submitted manuscript. Susceptibility to what? It would be better for this paragraph to do more to introduce the 'holes' in our understanding of the intestinal microbiota as not only prokaryotes but also eukaryotes (as introduced in lines 66-67). 

-Line 76 would be more accurate to state "increased diversity of bacteria in the presence of intestinal Blastocystis carriage." The prior sentences appropriately explain the questionable pathogenicity of this parasite, so the designation of 'infected gut' is therefore controversial. 

-Please do not use the terms 'flora' and 'microbiota' and 'microbiome'. Suggest changing 'flora' to 'microbiota' for consistency. Otherwise it is not clear what the difference is between flora and microbiota and whether these terms are synonymous or meant to designate something different to the reader.

-The references to Giardia in line 86 are too restricted. Please include references to other studies including Bartelt et al PLoSPathogens, 2017; work by Andre Buret laboratory and work by Scott Dawson/Steven Singer group. For E histolytica see papers by Stacey Burgess.

PLOS authors have the option to publish the peer review history of their article (what does this mean?). If published, this will include your full peer review and any attached files.

Reviewer #1: Yes: Amir Abdoli

Reviewer #2: No
---

## [Decision Letter · Decision Letter 1]

29 Nov 2022

Dear Dr Rajamanikam,

Thank you very much for submitting your manuscript "A harmless organism in the gut can be triggered to be harmful" for consideration at PLOS Neglected Tropical Diseases. As with all papers reviewed by the journal, your manuscript was reviewed by members of the editorial board and by several independent reviewers. The reviewers appreciated the attention to an important topic. Based on the reviews, we are likely to accept this manuscript for publication, providing that you modify the manuscript according to the review recommendations. 

Thank-you for the resubmission of your manuscript to PLOS Neglected Tropical Diseases. Two reviewers have carefully evaluated your manuscript, and reviewer 2 has noted some language that requires clarification surrounding the nature of your controls, and various other places to make the manuscript more easily understandable to a reader.

Sincerely,

Matthew Brian Rogers, Ph.D.

Academic Editor

Shan Lv

Section Editor

Thank-you for the resubmission of your manuscript to PLOS Neglected Tropical Diseases. Two reviewers have carefully evaluated your manuscript, and reviewer 2 has noted some language that requires clarification surrounding the nature of your controls, and various other places to make the manuscript more easily understandable to a reader.

Reviewer's Responses to Questions

**Key Review Criteria Required for Acceptance?**

**Methods**

-Are the objectives of the study clearly articulated with a clear testable hypothesis stated?

-Is the study design appropriate to address the stated objectives?

-Is the population clearly described and appropriate for the hypothesis being tested?

-Is the sample size sufficient to ensure adequate power to address the hypothesis being tested?

-Were correct statistical analysis used to support conclusions?

-Are there concerns about ethical or regulatory requirements being met?

Reviewer #1: (No Response)

Reviewer #2: Important clarifications have been made. There are still some apparent inconsistencies to resolve, for example "control" in Figure 6A and 6B don't appear to refer to the same thing, and that is confusing and requires significant work by the reader to comprehend (these concepts are complex).

**Results**

-Does the analysis presented match the analysis plan?

-Are the results clearly and completely presented?

-Are the figures (Tables, Images) of sufficient quality for clarity?

Reviewer #1: (No Response)

Reviewer #2: Regarding comment #4/6. To clarify, these questions addressed how the authors knew that the protease activity (figure 6A) was due to Blastocystis proteases and not bacterial-derived proteases that are different between different bacterial communities. Part of the confusion for this reviewer was that the terms "control" and "bacterial co-culture" in Figure 6 are not well defined elsewhere in the manuscripts. Based on the methods (new line 239), it is now clearer that 'control' means 'minimalized bacteria' where as 'bacterial co-culture' means the the non-bacterial minimalzied/standard xenic culture. From a different perspective, the experimental group is therefore the minimalized bacterial group if blastocysts culture normally includes the autochthonous xenic bacteria (conversely, in Figure 6B it appears that control indeed does refer to autochthonous bacterial culture). Line 239 helps to alleviate concerns that the bacterial proteases account for differences in Figure 6A. I'd suggest re-labeling Figure 6A as following "Bacterial co-cultured" should read "autochthonous culture" and "control" should read "minimalized bacteria culture". This makes more clear what was done and what is being compared.

Line 383: This paragraph needs some editing for grammar and clarity. It is still hard to tell what is going on. First, include the term colonic cell proliferation so it is clear what is being measured. For example, change the opening clause to read "Regarding the ability of Blastocystis antigens to promote colonic cell proliferation,..." instead of "When tested on the ability to proliferate cells". 

Line 384, should 'individual' be 'individuals'? Also, should the comma be removed? The authors are meaning antigens from Blastocystis isolated from symptomatic individuals that were later cultured with the bacteria from asymptomatic individuals. Correct? 

Line 384, 'bacteria from asymptomatic INDIVIDUALS' not bacteria from asymptomatic isolates

Line 385-386 should this read "Nonetheless, [?ANTIGENS from?] Blastocystis sp isolated from asymptomatic individuals that were cultured with bacteria from symptomatic individuals produced greater proliferation than antigens from Blastocysits isolated from asymptomatic individuals in autochonthonus culture?" As written in line 387 it isn't explicit what comparison was made. Greater than what?

**Conclusions**

-Are the conclusions supported by the data presented?

-Are the limitations of analysis clearly described?

-Do the authors discuss how these data can be helpful to advance our understanding of the topic under study?

-Is public health relevance addressed?

Reviewer #1: (No Response)

Reviewer #2: The toned down conclusion is much better and appropriate to the data presented.

However, new paragraph, Lines 542-550 needs revision. In Line 545-546, again the findings are associative and not causal. Instead of 'due to' the authors should state "associated with" ... a single subtype. Also, the only other pathogens screened were other parasites (not other bacteria or viruses) and the parasite diagnostics used were low sensitivity non-molecular based methods. So it is overstating to say that Blastocystis was a single infective agent. The entire clause "which is also a single infective agent" should be removed. Please add the limitation of not use molecular diagnostics and that therefore a) Blastocystis-free group may have had individuals with Blastocystis that wasn't cultivatable by the authors methods and that b) Blastocystis quantitative burden in stool cannot be compared between the two symptomatic groups.

**Editorial and Data Presentation Modifications?**

Reviewer #1: (No Response)

Reviewer #2: (No Response)

**Summary and General Comments**

Reviewer #1: (No Response)

Reviewer #2: The authors have provided a detailed response to most of the critiques. Some responses were misinterpreted, but they are minor and need not be further addressed. (eg. by Blastocystis status I did not mean differentiating symptomatic from asymptomatic, but rather presence or absence of Blastocystis. In this paper they used culture-recovery methods rather than molecular methods to assign presence or absence of Blastocystis. Future studies should incorporate qPCR if possible--https://journals.asm.org/doi/10.1128/JCM.01392-10). Overall I think the findings are of interest, appropriate for PLoSNTDs, but a few more revisions are needed before this is publication-ready.

In addition to specific section comments above, please pay careful attention to semantics in the paper. As in line 209, Line 217 should read "Three-day-old Blastocystis sp. cells from symptomatic individuals". The Blastocystis cells are not inherently symptomatic or asymptomatic. This same problem is present in Figure 6 legend where there is reference to 'symptomatic' and "asymptomatic" parasite culture. It happens again in lines 357-359 where line 357 reads "symptomatic individuals' whereas line 358 reads "asymptomatic isolates". Line 376 should read "Isolates from asymptomatic individuals".Line 378 should read "bacterial suspension from symptomatic individuals". Please correct throughout the manuscript. For this reason I recommended additional copyediting below.

Finally, and now that clarifications have allowed me to better comprehend this study, I suggest a change in the title. The authors don't show 'harm' from Blastocystis in the in vitro models and likewise it is not possible from the data presented to assign the organism as 'harmless'. A more appropriate title would be "Gut bacteria influence Blastocystis phenotypes and may trigger pathogenicity"

PLOS authors have the option to publish the peer review history of their article (what does this mean?). If published, this will include your full peer review and any attached files.

Reviewer #1: No

Reviewer #2: No

Figure Files:

Data Requirements:

Reproducibility:

References

---

## [Editor Report · Decision Letter 2]

14 Feb 2023

Dear Dr. Rajamanikam,

We are pleased to inform you that your manuscript 'Gut bacteria influence Blastocystis sp. phenotypes and may trigger pathogenicity' has been provisionally accepted for publication in PLOS Neglected Tropical Diseases.

Best regards,

Matthew Brian Rogers, Ph.D.

Academic Editor

Shan Lv

%CORR_ED_EDITOR_ROLE%

Thank-you for the re-submission of your manuscript, and apologies for the delay in returning this decision. After consideration by two reviewers we have made the decision that your manuscript is acceptable in it's current form for publication.

---

## [Editor Report · Acceptance letter]

13 Mar 2023

Dear Dr. Rajamanikam,

We are delighted to inform you that your manuscript, "Gut bacteria influence Blastocystis sp. phenotypes and may trigger pathogenicity," has been formally accepted for publication in PLOS Neglected Tropical Diseases.

Best regards,

Shaden Kamhawi

co-Editor-in-Chief

Paul Brindley

co-Editor-in-Chief
